# Resistance Circuit Training or Walking Training: Which Program Improves Muscle Strength and Functional Autonomy More in Older Women?

**DOI:** 10.3390/ijerph19148828

**Published:** 2022-07-20

**Authors:** Ayrton Moraes Ramos, Pablo Jorge Marcos-Pardo, Rodrigo Gomes de Souza Vale, Lucio Marques Vieira-Souza, Bruno de Freitas Camilo, Estélio Henrique Martin-Dantas

**Affiliations:** 1Federal Institute of Pará—IFPA, Campus Paragominas, Paragominas 68629-020, Brazil; ayrton.moraes@ifpa.edu.br; 2CERNEP, Research Centre, SPORT Research Group (CTS-1024), University of Almería, 04120 Almería, Spain; 3Department of Education, Faculty of Education Sciences, University of Almería, 04120 Almería, Spain; 4Active Aging, Exercise and Health/HEALTHY-AGE Network, Consejo Superior de Deportes (CSD), Ministry of Culture and Sport of Spain, 28040 Madrid, Spain; rodrigogsvale@gmail.com (R.G.d.S.V.); estelio.dantas@unirio.br (E.H.M.-D.); 5Postgraduate Program in Exercise and Sport Sciences, Institute of Physical Education and Sports (PPCEE), Rio de Janeiro State University (UERJ), Rio de Janeiro 20550-013, Brazil; 6Physical Education Course, State University of Minas Gerais, Passos 37903-204, Brazil; profedf.luciomarkes@gmail.com (L.M.V.-S.); brunodefreitascamilo@yahoo.com.br (B.d.F.C.); 7Doctor’s Degree Program in Nursing and Biosciences—PPgEnfBio, Federal University of the State of Rio de Janeiro—UNIRIO, Rio de Janeiro 21941-901, Brazil

**Keywords:** quality of life, functional independence, physical exercise

## Abstract

To evaluate the effects of two programs (resistance and walking training) on the functional autonomy and muscle strength (isometric and dynamic) of older women, 67 subjects were divided randomly into three groups: resistance training (RTG; Mean = 64.70 ± 6.74 years), walking (WG, Mean = 65.56 ± 7.82 years), and control (CG; Mean = 64.81 ± 4.34). The experimental groups underwent a 16-week intervention. Muscle strength (isometric and dynamic) and functional autonomy were assessed. The subjects participating in the RTG showed improvements in the comparison pre to post-test in the maximal forces of upper limb (MULS) (Δ% = 49.48%; *p* = 0.001) and lower limb (MLLS) (Δ% = 56.70%; *p* = 0.001), isometric biceps forces (BIS) (Δ% = 30.13%; *p* = 0.001) and quadriceps forces (QIS) (Δ% = 65.92%; *p* = 0.001), and in the general index (GI) of functional autonomy (Δ% = −18.32%; *p* = 0.002). The WG improved in all functional autonomy tests, except for the standing up from prone position test (SVDP). In strength tests, the WG obtained improvements only in the QIS (Δ% = 41.80%; *p* = 0.001) and MLLS (Δ% = 49.13%; *p* = 0.001) tests. The RTG obtained better results (*p* < 0.05) when compared to the WG and CG. The results allow us to infer that resistance exercise programs are more effective in increasing strength and functional autonomy, a fact that may mitigate the deleterious effects on health of aging.

## 1. Introduction

Human aging involves multiple factors and occurs in a unique and diverse way in each individual [1]. In Brazil, changes in the population demographic profile are evidenced, mainly, by the increase in the number of older people [2]. At this juncture, it becomes necessary to encourage active, healthy, and enjoyable aging in order to reduce the health problems common in individuals with advanced age [3].

The reduction in total muscle strength is common in the aging process and can compromise the performance of activities of daily living (ADL), impacting the mobility and functional autonomy of the older people [4]. Functional autonomy is related to physical independence and the ability to perform everyday tasks [5]. In this sense, the deleterious effects of aging on autonomy can generate a decrease in the performance of ADL-related skills and the gradual reduction of muscle functions [2]. With aging, older individuals can be affected by sarcopenia, which is a syndrome characterized by the progressive loss of muscle mass [6] that is directly related to mechanisms involving the remodeling of motor units, hormonal reduction, and protein synthesis, leading to decreased muscle strength (FM) with consequent functional dependence [7]. The loss of muscle mass in the lower limbs results in reduced walking speed, greater frailty, and fatigue with advancing age, negatively impacting the health of the older people [4]. In addition, there is an increase in oxidative stress and in a way that all its deleterious effects are combined, which can result in the appearance of serious pathophysiological disorders [8].

In order to reduce the deleterious effects of aging, physical exercise can mitigate the loss of muscle strength, promote joint mobility, and increase muscle tone, in addition to providing better performance in ADLs, and consequently, improving well-being, self-confidence, and functional autonomy, besides being an excellent strategy for the treatment of depression [6]. Physical exercise in conjunction with nutrition activates molecular mechanisms that are important for adaptive physiological responses in muscle function that can positively interfere with activities of daily living and prevent the deleterious effects of aging, reducing the state of stress [9,10]. Walking is an activity of daily living that is very frequently investigated in the literature, but is a well-planned and prescribed physical activity program of walking more effective in slowing the decline in functional capacity and muscle strength than a resistance training program is? These are questions that are addressed in this research. In this light, resistance training has been widely recommended to mitigate neuromuscular declines and promote favorable adaptations to health and quality of life in older women [11]. In addition, resistance training in older people with no previous experience allows to increase the capacity of maximal force production and the speed of force development [8].

Thus, the objective of this study was to compare the effects of two programs, one of resistance training and the other of walking, on the functional autonomy and muscle strength (isometric and dynamic) of older women. The hypothesis was that the RTG group should be more effective and improve more isometric and dynamic strength values and functional autonomy compared to the other group.

## 2. Methods

### 2.1. Participants

The study is characterized as experimental research. The research consisted of older women living in the city of Aracaju, Brazil and was assisted by the “Unidades Básicas de Saúde” (UBSs) Augusto Franco and Antônio Alves. Both UBSs were visited in order to publicize the research and start the sampling process. The present study was directed to older women.

The inclusion criteria were (i) being 60 years of age or older; (ii) being committed to participating in a physical exercise program and being independent in activities of daily living; (iii) not presenting cardiovascular comorbidities. Exclusion criteria were (i) participants who for some reason could not walk or had motor limitations; (ii) more than 20% absence during the intervention.

Sample size was calculated using the G*Power 3.1 software (Heinrich-Heine-Universität Düsseldorf, Düsseldorf, Germany) [12], using the following information: ANOVA with repeated measures for three groups and intra- and intergroup interaction, two measurement times, power of 0.95, α = 0.05, correlation coefficient of 0.5, correction for nonsphericity of 1, and an effect size of 0.25. The program estimated the sample size at 66 individuals as the minimum number of participants required for this investigation. It was verified that the sample size was sufficient to provide 80% of statistical power [13].

Due to a possible discontinuity in the intervention, a quantity approximately 10% larger than the estimated value was selected. Thus, after the sampling procedure, 76 individuals were selected. Then, the inclusion and exclusion criteria were applied after recruiting all 76 women. The older women were randomly divided by the Excel random function into three study groups, with 24 participants in each group. During the follow-up, five older women dropped out of the study due to a fall (1), family illness (2), and personal reasons (2). Thus, the final sample of this study consisted of 67 participants, divided into the resistance training group (RTG; *n* = 23), the walking group (WG; *n* = 22), and a control group (CG; *n*= 22). Figure 1 shows a flowchart diagram of randomization of the participants.

All study participants signed an informed consent form prior to inclusion in accordance with the guidelines regarding research on human subjects defined in Resolution 466/2012 of the National Health Council, and all study procedures were conducted following the principles of the Declaration of Helsinki. The study was approved by the Research Ethics Committee of the Tiradentes University with opinion number 3.936.886 CAAE n° 26524719.4.0000.5371.

### 2.2. Data Collection Procedures

#### 2.2.1. Anthropometric Measurements

To measure body mass and height, we used a mechanical scale with a capacity of 150 kg with precision of 100 g with a stadiometer of precision of 1 mm (Filizola^®^, Sao Paulo, Brazil). Subsequently, the body mass index (BMI) was calculated. The protocol of the International Society for the Advancement of Kinanthropometry was used as a parameter for anthropometric analysis [14].

##### Isometric Muscle Strength

To assess isometric muscle strength, a 300 kg digital force sensor (Chronojump^®^, Barcelona, Spain) was used on an evaluation platform. Two movements were evaluated: knee extension and elbow flexion.

To evaluate the knee extensor musculature, the older woman remained in a sitting position, with her trunk erect and her hands on the handles of the chair affixed to the evaluation platform. The knee joint remained at a 90-degree angle. The wristband affixed to the sensor cable was placed on the tibio-tarsal joint. At the command “go”, the participant performed knee extension [15].

To assess the elbow flexor musculature, the older woman stood on the strength assessment platform with her trunk erect, elbows glued to her trunk, and with semiflexion (10°–15°) of the elbow joint. The older woman held a steel bar with both supinated hands, which was connected to the platform’s force sensor by means of a steel chain. The various links of the chain allowed the angle of the joint to be varied [15,16]. Each movement was executed three times, through maximal voluntary isometric contractions performed for 5 s. The attempt that presented the highest peak force of the generated signal was computed.

##### Dynamic Muscle Strength

Dynamic muscle strength was assessed through the elbow flexion and extension tests for the upper limbs and the sit and stand test for the lower limbs [17]. In the elbow flexion and extension test, the participant, in the standing position, with feet shoulder-width apart and knee semibending, should hold a seven-kilogram (7 kg) barbell. She should, with arms extended, gripping the bar at a distance equal to or slightly wider than the shoulder width, and in supination, bring the bar to shoulder height, bending the elbows, then return to the starting position in a controlled manner for greater efficiency of movement, but at the highest possible speed.

Special attention must be given to the control of the final phase of forearm extension, in order to stabilize the upper arm, avoiding rocking movements of both the spine and the forearm, ensuring that complete flexion is performed (the evaluator may lightly hold the participant’s biceps).

It is important that the upper arm remains static during the test. The participant is encouraged to perform as many push-ups as possible within a time limit of 30 s, but always with controlled movements in both the flexion and extension phases.

The chair sit-up test begins with the participant sitting in the middle of the chair, spine straight and feet shoulder-width apart, fully supported on the floor. The chair should be approximately 43 cm long, with a backrest and no arms. A 7 kg bar is used to execute the movement. For safety reasons, the chair should be placed against a wall, or otherwise stabilized, to prevent it from moving during the test. One foot may be slightly forward of the other to help maintain balance. At the “start” signal, the participant rises to full extension (vertical position) and returns to the starting position. The chair is only for reference and safety, as the participant cannot sit down due to the need to maintain the cadence of the movement. The number of executions in 30 s is measured at the highest possible speed, with the use of a bar supported on the trapezius and not on the cervical vertebrae, holding the bar with the hand, maintaining a variable distance according to the characteristics of each participant, and bending the knees (squatting).

##### Functional Autonomy

Functional autonomy was assessed using a protocol developed by the Latin American Development Group for Maturity (GDLAM) [18,19,20]. This protocol consists of the following tests: walk 10 m (W10m) [21]; stand up from sitting position (SSP) five times, consecutively [22]; stand up from ventral decubitus position (SVDP) [23]; stand up from chair and move around the house (SCMH) [24]; put on and take off a t-shirt (PTS) [25]. The times of these tests were measured in seconds using a stopwatch (Casio, Brazil) in two attempts with a 5 min interval, considering the best result. The GI was calculated using the following formula.
GI = {[(W10m + SSP + SVDP+ PTS) × 2] + SCMH}/4
where W10m, SSP, SVDP, PTS, and SCMH are measured in seconds, and GI represents the GDLAM index in scores.

##### Intervention

After the initial two weeks, dedicated to the diagnostic evaluation, the groups were directed to their respective practices. In this phase, two weeks of adaptation and 16 weeks of intervention were performed. The American College of Sports Medicine recommendations were used for the prescription of combined resistance and strength training and cardiorespiratory training [26]. The training intensities and protocols were similar to those used in the study by Kukkonen-Harjula et al. [27] for cardiovascular training.

The RTG performed a circuit consisting of 11 strength exercises: biceps with barbell, extension chair, triceps on cross over, flexor chair, shoulder development with dumbbells, bench squat with the weight next to the chest, flyer, leg press, front pull, abdominal plank, and pelvic lift. The training frequency was twice a week, lasting 50 min per session, with three sets of 12–14 repetitions with two minutes rest between sets. This intervention was composed of (a) 10 min of warm up with submaximal stretching exercises and exercises with movements in the main joints; (b) 35 min of strength exercises, and (c) 5 min of muscle relaxation. Each resistance training session took place in the laboratory under the supervision of a researcher and sports physical educator.

The intensity of the muscular strength training effort was controlled by the OMNI-Resistance Exercise Scale (OMNIRES) [28]. In the first two weeks, a light-to-moderate level (level 3 to 5) was maintained, and in the following weeks, a moderate-to-intense level (level 6 to 8) was maintained. Regarding flexibility, the training was controlled by the perception of perceived effort, using the PERFLEX scale [29], keeping the stretching levels within the perception of effort from 31 to 60, and at the moment of the flexion performance between 61 and 80.

The WG did walking training 2 times a week for 16 weeks. The session duration was 50 min, divided into 3 parts: (a) 10 min of warm up with submaximal stretching and dynamic flexion exercises for the main joints; (b) 35 min of walking with effort control scores of 3–4 on the perceived exertion scale (Borg CR-10) [30] during the first two weeks, and scores of 5–6 in subsequent weeks, and (c) five minutes of muscle relaxation.

The control group (CG) participated in educational lectures and manual activities once a week throughout the study period.

### 2.3. Statistical Analysis

The data were presented as mean, standard deviation, and percentage variation. The normality of the data was verified by the Shapiro–Wilk test. Sphericity was analyzed with Bartlett’s test. Normality and sphericity were confirmed. ANOVA with repeated measures (3 × 2) was employed for inter- and intragroup comparison in the variables related to muscle strength and functional autonomy, followed by Bonferroni adjustment post hoc test. Cohen effect size (*d*) was calculated to analyze the clinical impact of the different interventions. It was used for interpretation: <0.2 (weak); 0.2–0.79 (moderate); ≥0.8 (strong) [31]. A *p-*value < 0.05 was adopted as statistically significant. Statistical treatment was performed by IBM SPSS Statistic 23 software (SPSS Inc., Armonk, NY, USA).

## 3. Results

At the beginning of the study, the groups did not present significant differences in the analyzed variables. Table 1 shows the characterization of the sample regarding anthropometric variables and age. It was observed that the groups presented a normal distribution in the variables analyzed according to the results of the Shapiro–Wilk test.

ANOVA with repeated measures showed an interaction between groups when comparing the moments before and after the intervention (F = 9.578; *p* < 0.001). In the isometric and dynamic strength tests (Table 2), there was an intragroup increase in strength in the variables biceps isometric strength (BIS) (*p* < 0.001), quadriceps isometric strength (QIS) (*p* < 0.001), maximum upper limb strength (MULS) (*p* < 0.001), and maximum lower limb strength (MLLS) (*p* < 0.001) for the RTG group. However, in the WG group, increases in strength were observed only in the QIS (*p* < 0.001) and MLLS (*p* < 0.001) variables. There was no change in muscle strength in the MULS (*p* = 0.158) and BIS (*p* = 0.189) tests in the WG. In the intergroup analysis, at the post-test moment, better performance of BIS (*p* < 0.001), QIS (*p* < 0.001), MULS (*p* < 0.001), and MLLS (*p* < 0.001) variables was observed in the RTG group compared to the WG group. The RTG scored better when compared to the CG in all muscle strength tests (*p* < 0.001). The WG was better than the CG in the QIS (*p* < 0.001) and MLLS (*p* < 0.001) strength tests.

Table 3 presents the comparative results of the functional autonomy tests obtained by applying the GDLAM protocol between the RTG and WG groups.

In the functional autonomy tests (Table 3), the RTG group had reduced execution time in the W10M (*p* = 0.002), SSP (*p* = 0.003), SVDP (*p* = 0.001), SCMH (*p* < 0.001), and PTS (*p* = 0.017) tests and in the GI scores (*p* = 0.002) in the intragroup evaluation.

Similarly, in the intragroup analysis, WG obtained faster times in W10M (*p* = 0.016), SSP (*p* = 0.013), SCMH (*p* = 0.017), PTS (*p* = 0.017), and GI (*p* = 0.002), but not in the SVDP test (*p* = 0.426). When performing the intergroup analysis, better results were observed for the RTG when compared to the WG in the SSP (*p* = 0.027), SVDP (*p* = 0.010), SCMH (*p* = 0.001), PTS (*p* = 0.011), and GI (*p* < 0.001) variables. No statistically significant differences were found between RTG and WG on the W10M test (*p* = 0.271). The RTG scored better when compared to the CG in all tests (*p* < 0.001) and in GI (*p* < 0.001). The WG was better than the CG in the W10m, SSP, and SCMH tests (*p* < 0.001) and in the GI (*p* < 0.001).

## 4. Discussion

The present study aims to evaluate the effects of two programs (RTG and WG) on functional autonomy and muscle strength (isometric and dynamic) in older women. The results of this study showed increases in isometric and dynamic strength, and a reduction in execution time of functional autonomy tests from pre to post-test in the RTG. The WG group obtained improvements in all functional autonomy tests, except for SVDP. Regarding the strength tests, the WG group performed better in the lower limb tests (QIS and MLLS). In the comparison between groups, it was found that resistance training resulted in greater changes when compared to the group that performed the walking program.

Similar results were observed in an investigation that analyzed the effect of resistance training on functional parameters in sedentary older women (68.4 years ± 5.2 years).Significant improvements were found in the 10 m walk test (6.5 ± 1.2 vs. 5.8 ± 1.1 s; *p* = 0.001), get up from sitting position (8.6 ± 2.3 vs. 6.9 ± 1.4 s; *p* = 0.001), getting up from a chair in 30 s (19.5 ± 5.0 vs. 23.8 ± 5.3 repetitions; *p* = 0.001), putting on and taking off a t-shirt (9.7 ± 1.8 vs. 9.0 ± 1.9 s; *p* = 0.025), and getting up from the prone position (3.6 ± 1.0 vs. 3.3 ± 0.9 s; *p* = 0.003) after 10 weeks of intervention [32].

In a study that analyzed 45 women aged 60 years or older, it was observed that after a 12-week resistance training intervention, the older women showed an improvement in muscle strength and functional fitness tests [33], corroborating the findings of this study. Moreover, in another study that analyzed older women who participated in a multicomponent training program (upper and lower limbs), it was possible to see, after 56 weeks, improvements in functional autonomy, mobility, and flexibility [34]. Similarly, in a study that evaluated 45 individuals aged 65–75 years, it was possible to find, after 12 weeks of intervention, that resistance training improved functional autonomy and increased upper and lower limb muscle strength [1].

Resistance training has been recommended because it is an important method that stimulates functional autonomy in older people [35], increasing independence and reducing the risk of mortality [36]. Evidence from the literature has already demonstrated the positive effects of resistance training on functionality, as in a systematic review that analyzed 12 studies and showed a positive effect on gait, balance, and walking speed in a straight line, allowing greater autonomy for the older people in performing activities of daily living [3]. Furthermore, a systematic review with meta-analysis involving 22 studies found that resistance training had a significant effect on muscle strength in older people aged 75 years and older [37]. Another meta-analysis indicated that greater gains in muscle strength, including upper body strength and functional abilities in older adults, can be achieved through mostly supervised gym-based programs [38]. Another recent study suggests that elastic band resistance training is an effective, portable, and cost-effective means of improving lower body function and muscle quality in an aging population, but that traditional free weight resistance training may be more impactful for whole body improvements [39]. Given the positive effects of resistance training on muscle strength and its relationship with independence, regular practice of resistance exercises would contribute to the maintenance/increase of functional autonomy in older women.

Regarding the effects of aerobic exercises on health, we noticed that the data on the relationship between walking, functional autonomy, and muscle strength in older women are still insipient, which reinforces the importance of the present investigation. In our study, improvement was observed in the aspects of functional autonomy and lower limb strength. In a pilot intervention study, performed through lateral walking for 6 weeks, it was possible to observe an improvement in walking speed and a reduction in the risk of falls in community-dwelling older people aged 65 years or older [40].

In another study, it was found that older people practicing walking showed better functional mobility when compared to sedentary older people [41]. Such findings partially corroborate the present investigation, from which it was observed that a walking program performed for 16 weeks, twice a week, was sufficient to improve the tests of functional autonomy (W10M; SSP; SCMH; PTS; GI) and dynamic strength (FIC and MLLS). It is important to highlight that aerobic exercises such as walking present greater stimulation to lower limb muscles, which justifies the results of the present investigation.

The decline in strength and muscle mass is common in the aging process and can be attenuated through the practice of aerobic and resistance exercises [36]. Resistance exercises can attenuate the effects of aging on neuromuscular functions by increasing the metabolic capacity of skeletal muscles, improving glucose homeostasis, and preventing intramuscular accumulation of lipids, which will contribute to improved physical performance [42]. In addition, aerobic training of walking but with Nordic walking poles did not show significant changes in upper body muscle strength in older women [43]. A recent study, which involved aerobic training with resistance poles and rubber bands, did show strength improvements in the upper body musculature in older women [44].

However, it is worth pointing out that although both exercises are beneficial to health, some types of aerobic exercise, such as walking, can target improvements specifically to the muscles that perform the movement. Thus, the performance of resistance or aerobic exercises that involve lower and upper limbs should be encouraged, since it could contribute, more broadly, to the increase/maintenance of muscle strength, impacting positively on functionality. This study is a novelty because there is no previous study evaluating the effects of these two training programs (RTG and WG) on dynamic and isometric muscle strength in the upper and lower limbs and on functional autonomy in older women.

This study has some limitations. The study sample was only women, and therefore the results cannot be extrapolated to the male population and to other age groups. From this perspective, research is needed on the participation of men and people belonging to different population groups. In addition, other studies with a larger sample size are recommended. It is important to highlight that the scarcity of studies that have analyzed two training protocols (walking and resistance training) makes it difficult to compare the findings of the present investigation with other studies. However, this reinforces the importance of this investigation in understanding the influence of different training protocols on functional autonomy and muscle strength in older women.

## 5. Conclusions

From this study, it was evident that the older participants of the RTG showed improvements in isometric and dynamic strength, and a reduction in the execution time of the functional autonomy tests when comparing pre and post-tests. The WG obtained improvements in all the functional autonomy tests, except for the SVDP. In strength tests, the WG obtained better performance only in the lower limb tests (QIS and MLLS). In the comparison between groups, it was found that the RTG resulted in more expressive changes when compared to the group that performed the walking program. The results of this study denote the importance of interventions with aerobic and/or anaerobic exercise programs that simultaneously involve the upper and lower limbs in order to stimulate the musculature globally, which could attenuate the deleterious effects to health due to the less active lifestyle of older individuals. The improvement in isometric and dynamic muscle strength contributes to improving the functional capacity linked to the activities of daily living of older people. This should be considered in the design of training programs for older people for a healthy aging. New studies are recommended to foster recommendations for improving autonomy and muscle strength in older adults.

## Figures and Tables

**Figure 1 ijerph-19-08828-f001:**
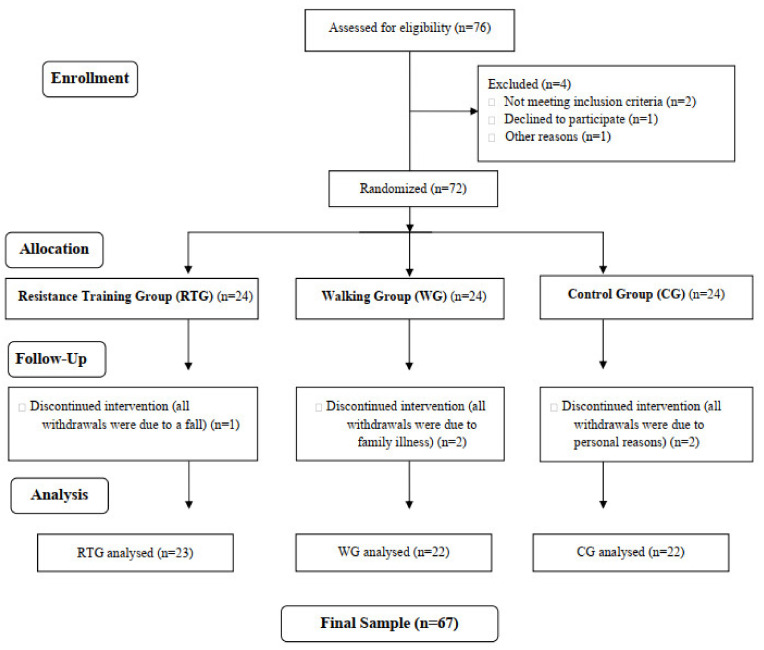
Consort 2010 flow diagram.

**Table 1 ijerph-19-08828-t001:** Characterization of the sample (*n* = 67).

Variables	Groups	Mean ± SD	*p*-Value (SW)
Age (years)	RTG	64.70 ± 6.74	0.321
	WG	65.56 ± 7.82	0.291
	CG	64.81 ± 4.34	0.435
Body mass (kg)	RTG	63.06 ± 11.01	0.154
	WG	68.94 ± 13.47	0.127
	CG	67.51 ± 6.57	0.173
Height (m)	RTG	1.53 ± 0.06	0.451
	WG	1.56 ± 0.07	0.428
	CG	1.58 ± 0.08	0.413
BMI (kg/m^2^)	RTG	26.88 ± 4.43	0.391
	WG	28.34 ± 4.72	0.349
	CG	27.18 ± 2.98	0.402

Legend: RTG = resistance training group; WG = walking group; CG = control group; SD = standard deviation; SW: Shapiro–Wilk test.

**Table 2 ijerph-19-08828-t002:** Comparative analysis of isometric and dynamic muscle strength tests between the study groups.

	Groups	Pre-Test	Pos-Test	Δ%	*d*
BIS (N)	RTG	154.24 ± 43.93	200.70 ± 27.81 *^,#,†^	30.13%	1.06 (s)
	WG	152.43 ± 37.63	160.89 ± 33.52	5.55%	0.22 (m)
	CG	151.31 ± 36.57	154.27 ± 34.61	2.0%	0.08 (w)
QIS (N)	RTG	223.75 ± 73.15	371.26 ± 57.81 *^,#,†^	65.92%	2.02 (s)
	WG	207.65 ± 77.31	294.44 ± 72.68 *^,†^	41.80%	1.12 (s)
	CG	202.72 ± 70.13	204.98 ± 67.76	1.1%	0.03 (w)
MULS (REP)	RTG	14.37 ± 4.21	21.48 ± 2.49 *^,#,†^	49.48%	1.69 (s)
	WG	13.59 ± 3.41	14.15 ± 2.61	4.09%	0.16 (w)
	CG	12.89 ± 3.68	13.17 ± 3.41	2.2%	0.07 (w)
MLLS (REP)	RTG	14.37 ± 3.34	22.52 ± 2.59 *^,#,†^	56.70%	2.44 (s)
	WG	12.81 ± 2.79	19.11 ± 1.65 *^,†^	49.13%	2.26 (s)
	CG	11.76 ± 3.02	12. 23 ± 2.58	4.0%	0.15 (w)

Legend: BIS: biceps isometric strength; QIS: quadriceps isometric strength; MULS: maximum upper limb strength; MLLS: maximum lower limb strength; * *p* < 0.05 (intragroup); ^#^ *p* < 0.05 RTG vs. WG; ^†^ *p* < 0.05 RTG or WG vs. CG. Δ%: percent change; *d*: effect size; (w): weak (*d* < 0.2); (m): moderate (0.2 ≤ *d* < 0.8); (s): strong (*d* ≥ 0.8). N: Newton; REP: repetitions.

**Table 3 ijerph-19-08828-t003:** Comparative analysis of the autonomy tests between the study groups.

	Groups	Pre-Test	Pos-Test	Δ%	*d*
W10M (s)	RTG	7.88 ± 3.15	6.57 ± 1.29 *^,†^	−16.63%	−0.42 (m)
	WG	7.95 ± 1.12	6.93 ± 1.05 *^,†^	−12.85%	−0.91 (s)
	CG	8.26 ± 2.70	8.41 ± 2.73	1.8%	0.05 (w)
SSP (s)	RTG	12.13 ± 3.08	10.47 ± 1.38 *^,#,†^	−13.68%	−0.54 (m)
	WG	13.06 ± 3.59	11.69 ± 2.40 *^,†^	−10.53	−0.38 (m)
	CG	12.93 ± 2.42	13.17 ± 2.88	1.9%	0.10 (w)
SVDP (s)	RTG	5.33 ± 2.67	3.91 ± 1.12 *^,#,†^	−26.67%	−0.53 (m)
	WG	6.48 ± 3.60	6.16 ± 4.22	−4.86%	−0.09 (w)
	CG	5.86 ± 1.67	6.37 ± 1.81	8.7%	0.30 (m)
SCMH (s)	RTG	47.37 ± 7.87	40.52 ± 4.11 *^,#,†^	−14.45%	−0.87 (s)
	WG	48.29 ± 5.86	45.46 ± 5.61 *^,†^	−5.87%	−0.48 (m)
	CG	48.91 ± 11.93	49.21 ± 12.55	0.6%	0.02 (w)
PTS (s)	RTG	15.11 ± 4.47	11.17 ± 1.69 *^,#,†^	−26.05%	−0.88 (s)
	WG	14.42 ± 3.63	12.66 ± 2.41 *	−12.20%	−0.48 (m)
	CG	14.44 ± 4.18	13.82 ± 3.87	−4.3%	−0.15 (w)
GI (scores)	RTG	32.07 ± 7.14	26.19 ± 2.08 *^,#,†^	−18.32%	−0.82 (s)
	WG	33.03 ± 5.73	30.08 ± 4.31 *^,†^	−8.91%	−0.51 (m)
	CG	32.96 ± 5.11	33.19 ± 5.15	0.7%	0.04 (w)

Legend: W10M: walk 10 m; SSP: get up from sitting position; SVDP: get up from ventral decubitus position; SCMH: get up from chair and move around the house; PTS: put on and take off T-shirt; GI: general index; * *p* < 0.05 (intragroup); ^#^ *p* < 0.05 RTG vs. WG; ^†^ *p* < 0.05 RTG or WG vs. CG Δ%: percent change; *d*: effect size; (w): weak (*d* < 0.2); (m): moderate (0.2 ≤ *d* < 0.8); (s): strong (*d* ≥ 0.8). Tests were measured in seconds (s), and the overall index in scores.

## Data Availability

The data presented in this study are available upon reasonable request from the corresponding author.

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
