# Peer review of "Resistance Circuit Training or Walking Training: Which Program Improves Muscle Strength and Functional Autonomy More in Older Women?"

_ijerph, 2022, doi:10.3390/ijerph19148828_

Round 1

Reviewer 1 Report

I received the paper entitled : Resistance circuit training or walking training: which program improves muscle strength and functional autonomy more in older women? 

After reading, I find this article well written and the methodology used is well explained and reproducible. The discussion of the results is well conducted. On the whole it is a good piece of work. However, I have two comments, one of which is major. 

- Table 1 : On the first line, I think it is 'SD' and not 'DP'.

- Lines 224 - 225 : The authors stated that they performed the Shapiro-Wilk and Bartlett tests. I would like the authors to say something about the results of these tests. This information will allow us to know whether the statistical tests used are justified. Also I would like to know why the authors chose the repeated measures Anova rather than the Manova since they have 3 groups with two measures for each group? 

Author Response

Reviewer 1:

I received the paper entitled : Resistance circuit training or walking training: which program improves muscle strength and functional autonomy more in older women? 

After reading, I find this article well written and the methodology used is well explained and reproducible. The discussion of the results is well conducted. On the whole it is a good piece of work. However, I have two comments, one of which is major. 

- Table 1 : On the first line, I think it is 'SD' and not 'DP'.

Response: Thank you very much for your suggestion, the change has been made.

- Lines 224 - 225 : The authors stated that they performed the Shapiro-Wilk and Bartlett tests. I would like the authors to say something about the results of these tests. This information will allow us to know whether the statistical tests used are justified. Also I would like to know why the authors chose the repeated measures Anova rather than the Manova since they have 3 groups with two measures for each group? 

Response: The results of Shapiro-Wilk are showed in Table 1. Normality and sphericity were confirmed. The change has been made in text. The variables showed normal distribution and are continuous. Thus, we followed the statistical treatment with a parametric approach (ANOVA with repeated measures) according to Triola (2022) (Triola, M. F. Introdução à estatísitica. 12a Ed. LTC editora, 2022)

Reviewer 2 Report

I have no further questions fo the authors

Author Response

Reviewer 2:

I have no further questions for the authors

Response: Thank you very much for your review

Reviewer 3 Report

The aim of this study was to compare the effect of two training programs (i.e., resistance training and walking training) on the functional autonomy and muscle strength (isometric and dynamic) of older women.

Congratulations to the authors for their work. They have developed a very interesting study with practical relevance for fitness professionals. In conclusion, the improvements in muscle strength and autonomy are superior in the strength training protocol (except for the walking test of GDLAM index) in comparison with the other training protocol. This provides very useful information since if physical-sports educators must choose a protocol to use with this population, they should select the latter. However, the writing of the manuscript as well as the discussion of the results needs to be improved to reflect the high quality of the study design.

Comments and Suggestions on each section of the document will follow.

TITLE

As for the title, I would suggest rephrasing it with a phrase indicating the most important result of the study.

ABSTRACT

In short, I would suggest using headers or writing it continuously.

I would not include in the summary how were the random or descriptive data on age by group (only the general data).

Unless I have misunderstood the results, I think there has been an abstract confusion between intra - and inter-group comparisons.I suggest that both appear in the abstract to answer the research question.

INTRODUCTION

In the introduction, I recommend structuring it in a less fragmented way.A paragraph must be composed of at least 3 sentences.

My suggestion: a first paragraph with the contextualization (figures of aging and description of the consequences and the need to develop active aging - line 44), a second paragraph where the idea raised in lines 61-64 and 71-75 is further developed.Finally, a paragraph arguing the purpose of the research, the objectives and the hypothesis (it is important that the literature mentioned above supports this hypothesis).In this regard, I recommend changing the verb "evaluate" to "compare ", as the ultimate goal is to see which protocol is most effective.It is therefore important to explain in the introduction the justification for this idea of comparing the two protocols.

METHODS

2.1. Study design and sample

I suggest renaming the subsection, as it only focuses on the sample.Participants or another synonym would be more appropriate.I also suggest writing this section more concisely and not fragmenting it so much into small paragraphs.Use connectors to make speech flow.Furthermore, with respect to the sample, it would perhaps be more appropriate to explain that the inclusion/exclusion criteria apply after recruiting the 76 women (or placing this part below).

The inclusion of the diagram (figure 1) provided by the authors for a better understanding of the text is greatly appreciated.

It should be noted in this section that participant characteristics are provided in the results section.

It is recommended to change the abbreviations of the groups to those appropriate for the English translation to facilitate the reading of the manuscript.In addition, all abbreviations throughout the manuscript must be revised to correspond to English and not Portuguese.

2.2. Data collection procedures

It is suggested that subsections be included for all variables (missing for "anthropometric measurements ").

The wording should be improved without mixing up inappropriate tenses and trying to synthesize the information more.

Some specific comments in this section would be:

Line 122: specify the device to measure the height.

Line 129: the device description can be omitted.

Line 132: it is suggested to include the description of each exercise after line 132 and conclude that paragraph with the way of measurement and selection of the attempts of both tests.Provide reference 17 (or others) in each of the procedure descriptions.

Lines 149 and 163: it is not necessary to specify that it is for women because the entire sample is of women.

Line 154: include in the previous paragraph

Line 158: include in the previous paragraph

Line 166.Include in the previous paragraph

Line 169.Include in the previous paragraph

(these last 4 Suggestions are to improve the writing and avoid fragmentation of the manuscript).

Line 182: the abbreviation gi is missing.Please remember to adapt the abbreviations to the language of the document.

As regards the intervention section:

A figure with descriptive intervention procedures would be advisable.

Line 196.Include it in the relevant paragraph of this protocol (214).

Line 207. Include in the previous paragraph.

Line 221. This last sentence should be removed to avoid misinterpretation.

2.3. Statistical analysis

Include the information about the statistical analysis performed for the descriptive variables and indicate those variables that were analyzed with ANOVA.

Since the sample size is different between groups, Hedges' g should be recommended.

Line 226. Change the order of intra- and inter-group or the 3x2 in order to have correspondence to the ANOVA factors.

RESULTS

For the descriptive results of the sample, it would be advisable to state in the text that there are no significant differences for any of these variables and then refer to the tables for the p-values.

Table 1: change the abbreviation DP (Desvio Padrão) to SD and remove the word “legend” in the table footnote.

For the ANOVA for the muscle strength and functional autonomy test variables, the main effects of each factor as well as the interaction should be reported (it is suggested that this be mentioned in the text). Subsequently, post-hoc results could be presented. This should be done for each variable to be analyzed.

Line 243: Improve the sentence formulation (lacks appropriate verb and adverb).

Tables 2 and 3: For a better understanding of the reader, also indicate the p-values in the table. To avoid repeating information, it is recommended that after describing the main effects and the interaction, describe which variables show differences and a range for the p-value. For example: There was an increase in the GTR group for FIB, FIC, FMMS, and FMMI (p ≥ 0.001).

Table 2 and 3: For †, to correct: p<0.05 GTR or GCAM "vs. GC"

Table 3. It is suggested to be consistent in all tables for the reporting of the units of the variables (at the side or at the bottom of the table).

For the presentation of the results, it is suggested to report them following the following steps (or similar) for each variable:

1. Main effects of time and group factors

2. Interaction between factors

3. If there is interaction, describe first the intra-group changes for each (refer also to whether there are changes in the CG, or indicate at the beginning of the results section - if applicable - that there are no changes for any variable in this group.

4. Describe the relationship of the protocols to the CG.

5. Comparison between protocols.

DISCUSSION

The discussion lacks an in-depth critical analysis of the results obtained and the contrast to existing ones in the literature. It should be improved the paragraphs that discuss the literature. The mere enumeration of results should be avoided. Authors should explain the potential reasons for the concordance/ discordance of these studies with their own, as well as the magnitude of the changes in their respective protocols.

Considering that there are systematic reviews and meta-analyses on the topic, it is suggested to highlight them previously in the introduction as a basis for the argumentation of the work.

It is recommended to write the discussion more fluidly, highlighting, on the one hand, the comparison (agreement and disagreement) with the existing literature for each training group, and, on the other hand, the comparison between them. Since there are not many studies comparing these two protocols, contrast with those studies (if any). If there are not, further reinforce the novelty of the study.

CONCLUSION

More summarized.

Line 358-359: Include in the limitations section as future contributions.

In summary, the drafting and structure of this manuscript should be reviewed. Nevertheless, it is important to highlight the quality of the study design.

Author Response

Reviewer 3:

The aim of this study was to compare the effect of two training programs (i.e., resistance training and walking training) on the functional autonomy and muscle strength (isometric and dynamic) of older women.

Congratulations to the authors for their work. They have developed a very interesting study with practical relevance for fitness professionals. In conclusion, the improvements in muscle strength and autonomy are superior in the strength training protocol (except for the walking test of GDLAM index) in comparison with the other training protocol. This provides very useful information since if physical-sports educators must choose a protocol to use with this population, they should select the latter. However, the writing of the manuscript as well as the discussion of the results needs to be improved to reflect the high quality of the study design.

Comments and Suggestions on each section of the document will follow.

TITLE

As for the title, I would suggest rephrasing it with a phrase indicating the most important result of the study.

Response: We thank the reviewer for his suggestion but even many training professionals think that older people get the same benefits from aerobic training. We want to awaken with this title the reader's interest in knowing the effects of both training programs in this population.

ABSTRACT

In short, I would suggest using headers or writing it continuously.

I would not include in the summary how were the random or descriptive data on age by group (only the general data).

Unless I have misunderstood the results, I think there has been an abstract confusion between intra - and inter-group comparisons.I suggest that both appear in the abstract to answer the research question.

Response: Thank you very much for your suggestion. The abstract has been changed..

INTRODUCTION

In the introduction, I recommend structuring it in a less fragmented way.A paragraph must be composed of at least 3 sentences.

My suggestion: a first paragraph with the contextualization (figures of aging and description of the consequences and the need to develop active aging - line 44), a second paragraph where the idea raised in lines 61-64 and 71-75 is further developed. Finally, a paragraph arguing the purpose of the research, the objectives, and the hypothesis (it is important that the literature mentioned above supports this hypothesis).In this regard, I recommend changing the verb "evaluate" to "compare ", as the ultimate goal is to see which protocol is most effective. 

Response: The reviewer's instructions have been followed and changes have been made.

It is therefore important to explain in the introduction the justification for this idea of comparing the two protocols.

Response: Thank you very much for your suggestion, the change has been made.

METHODS

2.1. Study design and sample

I suggest renaming the subsection, as it only focuses on the sample.Participants or another synonym would be more appropriate.

Response: Thank you very much for your suggestion, the change has been made.

I also suggest writing this section more concisely and not fragmenting it so much into small paragraphs.Use connectors to make speech flow.Furthermore, with respect to the sample, it would perhaps be more appropriate to explain that the inclusion/exclusion criteria apply after recruiting the 76 women (or placing this part below).

Response: Thank you very much for your suggestion, the change has been made.

The inclusion of the diagram (figure 1) provided by the authors for a better understanding of the text is greatly appreciated.

Response: Thank you

It should be noted in this section that participant characteristics are provided in the results section.

It is recommended to change the abbreviations of the groups to those appropriate for the English translation to facilitate the reading of the manuscript.In addition, all abbreviations throughout the manuscript must be revised to correspond to English and not Portuguese.

Response: Thank you very much for your suggestion, the suggestion has been made.

2.2. Data collection procedures

It is suggested that subsections be included for all variables (missing for "anthropometric measurements ").

Response: Thank you very much for your suggestion, the suggestion has been made.

The wording should be improved without mixing up inappropriate tenses and trying to synthesize the information more.

Some specific comments in this section would be:

Line 122: specify the device to measure the height.

Response: Thank you very much for your suggestion, the suggestion has been made.

Line 129: the device description can be omitted.

Response: Thank you very much for your suggestion. The text has been changed.

Line 132: it is suggested to include the description of each exercise after line 132 and conclude that paragraph with the way of measurement and selection of the attempts of both tests.Provide reference 17 (or others) in each of the procedure descriptions.

Response: Thank you very much for your suggestion, the suggestion has been made.

Lines 149 and 163: it is not necessary to specify that it is for women because the entire sample is of women.

Response: Thank you very much for your suggestion, the suggestion has been made.

Line 154: include in the previous paragraph

 Ok

Line 158: include in the previous paragraph

Ok

Line 166.Include in the previous paragraph

Ok

Line 169.Include in the previous paragraph

 Ok

(these last 4 Suggestions are to improve the writing and avoid fragmentation of the manuscript).

Response: Thank you very much for your suggestion, the suggestion has been made.

Line 182: the abbreviation gi is missing. Please remember to adapt the abbreviations to the language of the document.

Response: Thank you very much for your suggestion, the suggestion has been made

As regards the intervention section:

A figure with descriptive intervention procedures would be advisable.

Line 196.Include it in the relevant paragraph of this protocol (214).

Line 207. Include in the previous paragraph.

Line 221. This last sentence should be removed to avoid misinterpretation.

Response: Thank you very much for your suggestion, the suggestion has been made.

2.3. Statistical analysis

Include the information about the statistical analysis performed for the descriptive variables and indicate those variables that were analyzed with ANOVA.

Response: Thank you very much for your suggestion, the suggestion has been made.

Since the sample size is different between groups, Hedges' g should be recommended.

Response: Thank you very much for your suggestion, but we chose the Cohen effect size because the group variables are continuous and normally distributed (Fröhlich et al. Outcome Effects and Effects Sizes in Sport Sciences. International Journal of Sports Science and Engineering, Vol. 03 (2009) No. 03, pp. 175-179; Thomas J.R., Nelson J.K, Silverman, S.J. Métodos de pesquisa em atividade física. 6a Ed. Artmed, 2012; Cohen, J. Statistical power analysis for the behavioral sciences. 2. ed. Hillsdale, NJ: Lawrence Erbaum, 1988)

Line 226. Change the order of intra- and inter-group or the 3x2 in order to have correspondence to the ANOVA factors.

Response: Thank you very much for your suggestion, the suggestion has been made.

RESULTS

For the descriptive results of the sample, it would be advisable to state in the text that there are no significant differences for any of these variables and then refer to the tables for the p-values.

Response: Thank you very much for your suggestion, the suggestion has been made.

Table 1: change the abbreviation DP (Desvio Padrão) to SD and remove the word “legend” in the table footnote.

Response: Thank you very much for your suggestion, the suggestion has been made.

For the ANOVA for the muscle strength and functional autonomy test variables, the main effects of each factor as well as the interaction should be reported (it is suggested that this be mentioned in the text). Subsequently, post-hoc results could be presented. This should be done for each variable to be analyzed.

Response: Thank you very much for your suggestion, the suggestion has been made. The post-hoc results are presented in text.

Line 243: Improve the sentence formulation (lacks appropriate verb and adverb).

Tables 2 and 3: For a better understanding of the reader, also indicate the p-values in the table. To avoid repeating information, it is recommended that after describing the main effects and the interaction, describe which variables show differences and a range for the p-value. For example: There was an increase in the GTR group for FIB, FIC, FMMS, and FMMI (p ≥ 0.001).

Response: The post-hoc results are presented in text.

Table 2 and 3: For †, to correct: p<0.05 GTR or GCAM "vs. GC"

Response: Thank you very much for your suggestion, the suggestion has been made.

Table 3. It is suggested to be consistent in all tables for the reporting of the units of the variables (at the side or at the bottom of the table).

Response: Thank you very much for your suggestion, the suggestion has been made.

For the presentation of the results, it is suggested to report them following the following steps (or similar) for each variable:

  1. Main effects of time and group factors
  2. Interaction between factors
  3. If there is interaction, describe first the intra-group changes for each (refer also to whether there are changes in the CG, or indicate at the beginning of the results section - if applicable - that there are no changes for any variable in this group.
  4. Describe the relationship of the protocols to the CG.
  5. Comparison between protocols.

 Response: Thank you very much for your suggestion, It has been revised and adapted for better understanding.

DISCUSSION

The discussion lacks an in-depth critical analysis of the results obtained and the contrast to existing ones in the literature. It should be improved the paragraphs that discuss the literature. The mere enumeration of results should be avoided. Authors should explain the potential reasons for the concordance/ discordance of these studies with their own, as well as the magnitude of the changes in their respective protocols.

Considering that there are systematic reviews and meta-analyses on the topic, it is suggested to highlight them previously in the introduction as a basis for the argumentation of the work.

It is recommended to write the discussion more fluidly, highlighting, on the one hand, the comparison (agreement and disagreement) with the existing literature for each training group, and, on the other hand, the comparison between them. Since there are not many studies comparing these two protocols, contrast with those studies (if any). If there are not, further reinforce the novelty of the study.

Response: Thank you very much for your suggestion, It has been reviewed and your suggestions have been taken into account.

‘CONCLUSION

More summarized.

Response: Thank you very much for your suggestion, the suggestion has been made.

Line 358-359: Include in the limitations section as future contributions.

Response: Thank you very much for your suggestion, the suggestion has been made (line 971).

In summary, the drafting and structure of this manuscript should be reviewed. Nevertheless, it is important to highlight the quality of the study design.

Response: Thank you very much for your suggestion, the suggestion has been made.

Reviewer 4 Report

The authors compared the effects of resistance vs. walking training programs on functional performance and muscle strength in older women. The study was well conducted, with proper experimental design. However, there is no sufficient explanation (rationale) regarding the comparison between resistance and walking training. The comparison provided by the authors did not add a novel and relevant information to the research area. Why would anyone expect a different result found by the authors? Resistance training imposes higher physiological stress than walking exercise. Therefore, it is clear that a higher increase in muscle strength and functional performanc would be observed after resistance training compared to walking training.

Author Response

Reviewer 4:

The authors compared the effects of resistance vs. walking training programs on functional performance and muscle strength in older women. The study was well conducted, with proper experimental design. However, there is no sufficient explanation (rationale) regarding the comparison between resistance and walking training. The comparison provided by the authors did not add a novel and relevant information to the research area. Why would anyone expect a different result found by the authors? Resistance training imposes higher physiological stress than walking exercise. Therefore, it is clear that a higher increase in muscle strength and functional performanc would be observed after resistance training compared to walking training.

Response: We agree with the reviewer but there are still trainers who think that older people should not or cannot do strength training. There are many trainers and even doctors who tell older people to go for a walk. In this study, we wanted to show that walking is good but that resistance training adapted to the characteristics of older people is much better. Still many people think that walking is sufficient in this age group. There is little current evidence in experimental studies to prove it, so we thank the reviewer for his comment and we want him to understand why we conducted this study. When it comes to recommending that older people do physical exercise, it is much better for them to do resistance training than just going for a walk. As we can see it improves their dynamic and isometric strength and functionality which influences their activities of daily living.

Furthermore, is included in the discussion “This study is a novelty because there is no previous study evaluating the effects of these two training programs on dynamic and isometric muscle strength in the upper and lower hemisphere and on functional autonomy in older women”.

Reviewer 5 Report

This is an overall interesting study about the effects of circuit training versus walking on muscle strength.

The design of the study is sound, and one strength is the additional focus on functional autonomy and activities of daily living, which allows for a more real-life oriented discussion of the results.

The main results of the study on muscle strength however, are not new: it is not surprising, that walking training has primarily an effect on the lower limbs and that a full-body training has also effects on the upper limbs, too.

In this context it could also be awaited, that those functional tests, which also comprised upper-body performance showed a higher benefit after GTR compared to walking and controls.

But nevertheless it is worth to prove these effects of whole-body training, particular in daily activities.

Points that need revision:

Introduction, ll58-59 and ll 65-68: here, the authors introduce the complex role of oxidative stress in the process of aging. This is indeed a very important aspect, but this focus has not been further addressed throughout the whole manuscript, and therefore I wonder, why at all this was mentioned here. I would recommend to delete the part about oxidative stress completely, as it does not further contribute to the study, since any parameters of oxidative stress were neither measured nor discussed later.

Ll 86-87: I do not understand why the sample was female because of a university project? Please, clarify or just state, that the study was focused on women.

Ll 112 ff: I do not clearly understand the reason why the study was not registered, please clarify. Or, since the study was approved by the Ethics committee, I would recommend to delete the sentence about registration to Clinical Trials.

Table 3: is IG GI? I don’t understand the measurement of the GI in “scores” From the table it seems to be seconds, and from the formula it should be seconds? Please, clarify.

Author Response

Reviewer 5:

This is an overall interesting study about the effects of circuit training versus walking on muscle strength.

The design of the study is sound, and one strength is the additional focus on functional autonomy and activities of daily living, which allows for a more real-life oriented discussion of the results.

The main results of the study on muscle strength however, are not new: it is not surprising, that walking training has primarily an effect on the lower limbs and that a full-body training has also effects on the upper limbs, too.

In this context it could also be awaited, that those functional tests, which also comprised upper-body performance showed a higher benefit after GTR compared to walking and controls.

But nevertheless it is worth to prove these effects of whole-body training, particular in daily activities.

Points that need revision:

Introduction, ll58-59 and ll 65-68: here, the authors introduce the complex role of oxidative stress in the process of aging. This is indeed a very important aspect, but this focus has not been further addressed throughout the whole manuscript, and therefore I wonder, why at all this was mentioned here. I would recommend to delete the part about oxidative stress completely, as it does not further contribute to the study, since any parameters of oxidative stress were neither measured nor discussed later.

Response: Thank you very much for your suggestion. The text has been changed.

Ll 86-87: I do not understand why the sample was female because of a university project? Please, clarify or just state, that the study was focused on women.

Response: Thank you very much for your suggestion, the suggestion has been made.

Ll 112 ff: I do not clearly understand the reason why the study was not registered, please clarify. Or, since the study was approved by the Ethics committee, I would recommend to delete the sentence about registration to Clinical Trials.

Response: Thank you very much for your suggestion, the suggestion has been made.

Table 3: is IG GI? I don’t understand the measurement of the GI in “scores” From the table it seems to be seconds, and from the formula it should be seconds? Please, clarify.

Response: Thank you very much for your comment. The GI value is in scores according to the GDLAM protocol validation process.

Round 2

Reviewer 3 Report

The revised version essentially addresses all the points I raised.  The authors have done a good job of synthesising all the findings. 

Just to note that the abbreviation GI has not been introduced anywhere in the manuscript (only in the abstract).

Congratulations to the authors for the study and the work they have done.

This manuscript is a resubmission of an earlier submission. The following is a list of the peer review reports and author responses from that submission.

Round 1

Reviewer 1 Report

The manuscript entitled "Resistance circuit training or walking training: which program improves muscle strength and functional autonomy more in older women?" it's an interesting job.

It is now known that adapted physical activity helps aging and at the same time is recommended in the case of numerous diseases such as diabetes, hypertension, cancer and neurocognitive disorders.

However, it is also well known that intense physical activity can instead represent a substrate for the appearance of certain disorders such as: infections of the skin, urinary tract or respiratory tract.

In light of this, both in the introduction and in the discussions it would be useful to underline these differences and if there are similar works, it would also be appropriate to know the state of health of the women under study (age of menopause, drug therapies, hypertension, nutrition) to understand if further stratification was possible.

In addition, he knows if there are similar studies carried out on male populations and finally underline the differences with publications on physical activity works.

I suggest some readings about physical activity and nutrition and the importance of physical activity in case of infections ( doi: 10.3390 / ijerph17249424).

Reviewer 2 Report

This study is a very significant randomized controlled trial that compared the effects between resistance training and walking for muscle strength and functional autonomy. However, I believe that there is a lack of content that should be complied with when reporting the results of randomized controlled trials. I had the following comments requiring clarification and suggestions to improve the manuscript.

  1. If the protocol has been registered in the International clinical trials register, please indicate the registration number. Authors are strongly encouraged to pre-register clinical trials with the international clinical trials register and cite a reference to the registration in the Methods section. If authors had not registered their trial they should explicitly state this and give the reason. Many journals specifically stated that all recent clinical trials must be registered as a requirement of submission to that journal.
  1. I recommend that randomized controlled trial be reported following CONSORT (Consolidated Standards of Reporting Trials). MDPI requires a completed CONSORT 2010 checklist and flow diagram as a condition of submission when reporting the results of a randomized trial.
  1. I have the impression that the scientific background and explanation of rationale on which this study was designed is insufficient. Please describe the novelty, interest, relevance, and significance of this study based on scientific evidence in the introduction, using the systematic review and other scientific evidence described in the discussion section.
  1. Of course, although it is important to note that this is the first study of its kind in Brazil, the references are too many from Brazil. I believe that referring to an international, high evidence level study would make the arguments more reliable.
  1. What is the main outcome of this study? Which indicator was used to set the effect size? What basis were they set? By presenting the main outcome and secondary outcomes of this study, the program outline and its effectiveness will be clearer.

Reviewer 3 Report

In this study authors compared  the effects of two programs, (resistance training vs aerobic), on functional autonomy and muscle strength  of  older women. Overall the study is of general interest iad well conceived

I have some observations:

Introduction: In my opinion authors should clearly specify what was the primary endpoint of the present study and what were secondary endpoints

Exclusion critera: there is no mention about health state of the over 60 years old subjects. In particular authors do not give informations about  cardiovascular diseases or cardiovascular disease risk factors among participants : were they ruled out before the entollement?

In the method session authors describe how they calculated tha sample size: please specify  the expected effect size 

Results: I suggest the addition of one or two figures (at least for the primary endpoint) in order to improve results presentation

Discussion: Authors should underline which are novelties  what are the novelties brought by present  study. It is not surprising that the GTR group presents a greater increase of muscle strength compared to GCAM  

Reviewer 4 Report

I have read with great attention the present article which deals with the theme of the quality of life of the older adults. I have therefore noted a few remarks that will enable the authors to improve the quality of their work. I would like the authors to consider these remarks in order to provide clear and thorough answers.  

  • Why did the authors choose to conduct the study only in women? There is a lack of justification for this.
  • The acronym GDLAM should definitely be avoided in the abstract. 
  • Line 67 'But is walking really the best physical activity program for older people?
  • I think the authors before asking this question need to do a paragraph on studies that have looked at the effects of walking and resistance training on their dependent variables. 
  • In the discussion, the authors discussed their results but did not address the physiological mechanisms that may explain the observed improvements based on the program offered to participants. I would like the authors to address this.
  • Finally, the new avenues of research proposed by the authors are terse and I would like to see the authors expand on future avenues of research.